# Immunopathogenesis of Nipah Virus Infection and Associated Immune Responses

Brent Brown [1,*], Tanya Gravier [2], Ingo Fricke [3], Suhaila A. Al-Sheboul [4], Theodor-Nicolae Carp [5], Chiuan Yee Leow [6], Chinua Imarogbe [7] and Javad Arabpour [8]

1   Biochem123 Ltd., London NW7 4AU, UK
2   Independent Researcher, San Francisco, CA 94102, USA; tanya.j199@icloud.com
3   Independent Immunologist and Researcher, Imkers Brink, 31195 Lamspringe, Germany; ingo.fricke@protonmail.com
4   Department of Medical Laboratory Sciences, Faculty of Applied Medical Sciences, Jordan University of Science and Technology, Irbid 22110, Jordan; sashboul@just.edu.jo
5   Independent Researcher, 050322 Bucharest, Romania
6   School of Pharmaceutical Sciences, Universiti Sains Malaysia, Pulau Pinang 11800, Malaysia; yee.leow@usm.my
7   UKHSA, Rosalind Franklin Laboratory, Royal Leamington Spa CV31 3RG, UK; rogbe2005@outlook.com
8   Independent Researcher, Tehran 1949635881, Iran; bio.javad.arabpour@gmail.com
*   Correspondence: abrownbscmsc@gmail.com

**Abstract:** Pandemics in the last two centuries have been initiated by causal pathogens that include Severe Acute Coronavirus 2 (SARS-CoV-2) and Influenza (e.g., the H1N1 pandemic of 2009). The latter is considered to have initiated two prior pandemics in 1918 and 1977, known as the "Spanish Flu" and "Russian Flu", respectively. Here, we discuss other emerging infections that could be potential public health threats. These include *Henipaviruses,* which are members of the family *Paramyxoviridae* that infect bats and other mammals. *Paramyxoviridae* also include Parainfluenza and Mumps viruses (*Rubulavirus*) but also Respiratory Syncytial virus (RSV) (*Pneumovirus*). Additionally included is the Measles virus, recorded for the first time in writing in 1657 (*Morbillivirus*). In humans and animals, these may cause encephalitis or respiratory diseases. Recently, two more highly pathogenic class 4 viral pathogens emerged. These were named Hendra *Henipavirus* (HeV) and Nipah *Henipavirus* (NiV). Nipah virus is a negative-sense single-stranded ribonucleic acid ((−) ssRNA) virus within the family *Paramyxoviridae*. There are currently no known therapeutics or treatment regimens licensed as effective in humans, with development ongoing. Nipah virus is a lethal emerging zoonotic disease that has been neglected since its characterization in 1999 until recently. Nipah virus infection occurs predominantly in isolated regions of Malaysia, Bangladesh, and India in small outbreaks. Factors that affect animal–human disease transmission include viral mutation, direct contact, amplifying reservoirs, food, close contact, and host cell mutations. There are different strains of Nipah virus, and small outbreaks in humans limit known research and surveillance on this pathogen. The small size of outbreaks in rural areas is suggestive of low transmission. Person-to-person transmission may occur. The role that zoonotic (animal–human) or host immune system cellular factors perform therefore requires analysis. Mortality estimates for NiV infection range from 38–100% (averaging 58.2% in early 2019). It is therefore critical to outline treatments and prevention for NiV disease in future research. The final stages of the disease severely affect key organ systems, particularly the central nervous system and brain. Therefore, here we clarify the pathogenesis, biochemical mechanisms, and all research in context with known immune cell proteins and genetic factors.

**Keywords:** *Henipavirus*; Hendra; innate; adaptive; immunology; molecular; Nipah; Paramyxovirus; virus

## 1. Introduction

Newly emerging viral infections remain a potential public health threat, including *Henipaviruses* discovered in the 1990s. Two of these are highly pathogenic in humans and are detailed below. During pathological research, Koch (1884) defined four key measures of microorganisms (microbes). Firstly, microbes should be found in abundance in disease and not in health; secondly, a microbe should be isolated from a host and grown in culture. In addition, a microbe should cause disease when infecting a host organism and be reisolated from inoculated or affected hosts to isolate the causal agent.

Within the binomial classification of kingdoms, *Henipaviruses* are defined under the kingdom *Orthornavirae* and order *Mononegavirales,* that include (−) ssRNA viruses within the family *Paramyxoviridae* (see Figure 1) [1,2].

**Figure 1.** Historical structure perspectives of *Paramyxoviridae*.

In comparison, other viruses include (+) ssRNA viruses, with the differences being that (−) ssRNA viruses have complementary RNA and require transcription to produce (+) ssRNA to produce mRNA and then translation by RNA polymerase enzymes. Other categories include those of single- and double-stranded deoxyribonucleic viruses (ssDNA/dsDNA) [3,4]. Negative-sense ssRNA viruses historically include those that may cause unusual levels of disease or mortality in vertebrates. For example, Ebola virus, Hantavirus, Influenza, Lassa fever virus, and Rabies virus are only examined from tissue samples within Biosafety Level (BSL) 1–4 laboratories based on potential pathogen risk categorization (see Figure 1 and Supplementary Materials). Nipah virus is categorized as a BSL4 pathogen.

The homologous *Paramyxoviridae* Hendra virus was first described in 1994, when initial reports appeared of a *Morbillivirus* in horses in Australia with high mortality of unknown origin [5]. In humans, these can cause encephalitis and/or respiratory disease. Nipah virus is named after the location of initial outbreaks in Malaysia around late 1998. Further outbreaks in humans and animals, and/or more specifically in livestock, have

occurred in Australia, Malaysia, Singapore, Cambodia, and Bangladesh. Mortality has been seen with NiV infection in at least six species, including horses, cats, and dogs, as well as ferrets, hamsters, guinea pigs, and monkeys. Besides Malaysia/Singapore (1998–1999), other reports of NiV infection were documented in Bangladesh (2001/2003/2004). Similarly, outbreaks were also seen in India (2001/2007/2018), and more recently in 2021 [6]. Nipah virus is classified as a Biosafety Level 4 (BSL4) pathogen because of its high fatality rates, thought to be 35% or more, requiring further clarification. In the first outbreak (1998), within 265 individuals, it was indicated that fatality rates were around 40% initially. Data on Uniprot is suggestive that at least two of the following hosts have NiV infections: *Cynopterus brachyotis* (the lesser short-nosed fruit bat), *Eonycteris spelaea* (the lesser dawn bat), *Macroglossus spelaeus* (the cave nectar bat), *Homo sapiens* (the human), *Pteropus vampyrus* (the large flying fox bat), *Scotophilus kuhlii* (the lesser asiatic yellow bat), but was also dominant in *Sus scrofa* (the pig), with all viral genes seen in these species (see Supplementary Materials) [7].

Initial reports were indicative of Pteropid bats (flying foxes) and *P. hypomelanus* as hosts of NiV potentially replicating or mutating [8]. More recently, before 2012, due to information technology (IT) advancement, as well as culturing and isolating other *Henipaviruses*, this has subsequently occurred. For example, the Cedar virus (CeV) was isolated from Australian bat urine samples of *P. alecto* [9]. Concurrent surveillance during a 2004 human outbreak clarified this through serological analysis. It was indicated then that birds, pigs, dogs, shrews, and rodents did not show signs of prior infection with trace amounts of antibodies reactive against NiV antigens. Antibodies reacting against antigens were found in *P. giganteus* [10,11]. Subsequently, NiV was recovered from *P. hypomelanus* urine, and the hosts of the NiV infection could be considered bats. However, other species like pigs are thought to be amplifying viral hosts [10]. The NiV genome is known to be a (−) ssRNA virus approximated at 18 kb but has been indicated as variable in recent years between provisional genomic sequencing clades. Some reports indicate that mosquitoes could be vectors. There are currently around 1300 species of bat, with NiV transmission rates in humans unknown as measured by R0, although considered low. This may be largely due to comparative research developments alongside next generation sequencing (NGS), as determined by genomic surveillance. However, immune system regulation requires further evaluation, as described below. This regulation can vary between different host species expressing immune cell markers that include cluster of differentiation markers (CD), cytokines (e.g., interleukins, IL), chemokines (CXC), but also specifically interferons (IFN). Live NiV isolation continues to be difficult, with severity measured according to laboratory and clinical reports due to sporadic outbreaks of such virulence. Other relevant *Paramyxoviridae* also infect humans with varying virulence; these include Human Para–Influenza viruses (HPIV types I–V), Respiratory Syncytial virus (RSV), and Measles (MeV). These differentially affect age groups and have historically caused severe illness in younger children and other vulnerable groups. Recently in 2021, Para–Influenza virus was genetically characterised into two sub–clades (4A and 4B) [12]. According to current reports, the incidence of encephalitis since 2016 has been 1.5/100,000 people, regardless of pathology. This equates to 120,000 individuals per year globally. Predominant infections that can cause encephalitis are Herpes Simplex virus (HSV) and Varicella-Zoster virus (VZV). Other additional viruses that may cause encephalitis include Enteroviruses, Japanese Encephalitis virus, Tick-Borne Encephalitis virus, but also Flaviviruses, and Alphaviruses. Bacteria like Mycobacterium Tuberculosis and Listeria Monocytogenes can also cause encephalitis [13,14]. Therefore, the aim of this paper is to clarify current biochemical research within NiV and HeV viral diseases that does cause excessive mortality in comparison to other well-characterised viruses where immunogens are licensed. Utilisation of vaccines against the MeV and Mumps viruses in comparison has demonstrated a clear reduction in disease burden and mortality [15–17].

## 2. Epidemiological and Clinical Perspectives

### 2.1. Clinical Manifestations and Diagnosis

Nipah virus infection frequently causes severe disease and mortality, with symptoms that include encephalitis and excessively high infection fatality rates (IFR) like other BSL4 pathogens (e.g., Ebola (EBOV)). Sensory receptors for pathogens exist within all biological systems under investigation below with regards to NiV research. These receptors are expressed within the respiratory, nervous, circulatory, endocrine, reproductive, muscular, and skeletal systems. In our last paper, we examined systemic immune system cytokines, chemokines, and CD markers during SARS-CoV-2 infection in depth [17]. However, below we discuss the pathogenesis of NiV infection and immune system regulation.

Clinical symptoms may include headache, fever, and muscle aches, as well as fatigue. Muscle stiffness, confusion, agitation, seizure, speech, and hearing issues can occur. In children, nausea, irritability, poor feeding patterns, and rigor can be considered (see Supplementary Materials). Comparisons of NiV infections are made with Japanese encephalitis. Severe central nervous system (CNS) symptoms could be considered similar; however, this also includes the peripheral nervous system (PNS) and immune system regulation.

Early analysis indicated through brain magnetic resonance (MR) scans that the subcortical and deep white matter of the brain possessed bilateral foci. This was illustrated to potentially affect periventricular areas and the corpus callosum with critical lesions less than 1 cm in diameter that may differentiate NiV infection and Japanese encephalitis virus [18]. Other neurotropic viruses can be ruled out with cerebrospinal fluid (CSF) analysis indicative of lymphocyte infiltration (pleocytosis) [6]. Initial symptoms that can occur (4–14 days) are fever and headache, with concurrent signs of acquired respiratory distress syndrome (ARDS). Subsequent cough, sore throat, and difficulty breathing can also occur. Concerningly, compartmentalized brain tissue swelling (oedema) can occur from infection, trauma, or stroke. Encephalitis is considered inflammation derived from chronic viral or acute infection accompanied by drowsiness, disorientation, and mental confusion that can progress to a coma within 24–48 h [19]. Cell counts can clarify this further. Recent population studies clarify that this can vary according to age, country, season, viral genetic mutations, and immune status. At least two of the following symptoms were indicated in recent reports as being indicative of encephalitis: fever, seizures, or focal neurological findings. Concurrent brain parenchyma with CSF pleocytosis of more than four leukocytes per μL is suggested in recent reports [20–22]. Viral encephalitis can be characterised by a combination of systemic effects, including vasculitis and necrosis, in both the CNS and PNS [22]. Extensive cellular infection of neurons, endothelial cells, and smooth muscle cells of the vasculature can occur. Initial interstitial pneumonia, accompanied by syncytial cell formation, may be observed with mononuclear cellular infiltrates resembling other pathologies. Additionally, long-term sequelae in survivors may include encephalopathy, ocular motor palsy, cervical dystonia, and facial paralysis [23]. It is notable that the mean age of survivors was 14.5 years old in the prior report; this is therefore clearly indicative of severity in children with an infection similar to MeV. The Nipah virus can be described as a neurotropic virus. However, we examine the factors relevant to clarify and explain how T cell responses, a critical part of immune system development during childhood, can be considered further. Few reports describe the exact nature of adaptive T cell responses apart from those seen in Cytomegalovirus (CMV) and Respiratory Syncytial virus infections so far [24–26].

### 2.2. Human Transmission of Henipaviruses

Initial reports remain unclear about the routes of NiV transmission. This is likely to be a combination of oropharyngeal and nosocomial routes or through fomites. It was thought that NiV could be transmitted through the date palm until 2014, according to earlier outbreaks in Bangladesh [27]. Outbreaks in India (2001–2007) continued to illustrate the similar severity of NiV disease with high IFR occurrence (74–100%), but also recently in 2019 in Bangladesh with similar IFR [28]. Prior reports during the NiV outbreaks in

Bangladesh were suggestive of the duration of illness leading to mortality at <6 days. Literature indicates that there are few reports of successful live NiV cultures apart from those obtained from samples of *P. hypomelanus*, P. vampyrus, and P. lylei bats. However, researchers in Australia (2011) investigated other Pteroptids, like *P. alecto,* and concurred that these could be considered natural host reservoirs where *Henipaviruses* co-evolve. It was indicated in samples analyzed for HeV or NiV infection that symptoms in bats appear to be subclinical with a low serological response [29]. Reverse transcriptase polymerase chain reaction (RT–PCR) is often utilised with cycle threshold (Ct) values compared for HeV genomes from urine, blood, throat, and rectal swab samples. The duration of PCR detection from urine HeV swabs is indicated at 10–19 days following infection in animals. In 2014, NiV was isolated from *P. medius,* with evidence that NiV spillover is indeed rare [30]. Importantly, during the initial Bangladesh (2004) outbreaks, NiV viral RNA was successfully isolated, meeting the above definition of a microbe [31].

### 2.3. Animal Host Reservoirs of Henipaviruses

Early outbreaks of NiV infection and concerns about pig vectors resulted in the slaughter of 1 million pigs due to viral severity. Recent reports have examined bat novel virus transmission in different species, indicating that species like *Myotis* spp. rather than *Pteropid* spp. could be reservoir hosts. Also, in an early surveillance study (n = 692), protein analysis indicated the NiV proteins below could be immunogenic. In a cross-sectional study of bat species, it was seen that bat samples (n = 33) possessed antibodies that were not cross-reactive against cellular infection with either HeV or NiV [32]. In a single-case isolation study, NiV RNA was detected in blood, bronchoalveolar lavage fluid (BALF), and CSF alongside serum IgM antibodies in people. In variable bat species, the immunoglobulin G (IgG) specific for NiV was indicated to be found in *P. medius* (21%), and *R. leschenaultia* (37.73%). Furthermore, neutralising antibodies (nAb) evidence of NiV infection were found in *P. medius* (20.68%) that were anti–NiV immunoglobulin G (IgG) antibodies. Thus far, the genomic clades NiV-M and NiV-B, which refer to the initial outbreaks in Malaysia and Bangladesh, have been identified. Sequencing recently was indicative of slight variations in genome size (15,100–18,172 bps) in *Pteropus* bats, with NiV not found in *Rousettus* spp. during surveillance [33]. Serology reports in bat reservoirs are variable. In a recent outbreak case study, surveillance of bats (2018/2019) confirmed that 3.2% and 25% of the samples obtained were seropositive for NiV infection. This indicated that bat immune responses, as indicated by serology, could plausibly be established at cyclical levels until 6–7 years after NiV infection [34]. Indeed, in animal surveillance prior to 2011, reports did indicate that bats may have differential regulation of immune responses, with further research required to investigate this [29].

## 3. Henipavirus Pathogenesis

### 3.1. Introduction to Nipah Virus Proteins and Genes

Currently, the Uniprot database shows 281 proteins listed under *Henipavirus* that include nine from the original HeV 1994 strain (AF017149) and eight from NiV strains (UP000128950). Nipah virus is known to be approximately 18,200 bps in size and encode six proteins. These are nucleocapsid (N), phosphoprotein (P), matrix protein (M), fusion protein (F), glycoprotein (G), and polymerase (L) protein. An additional NiV P protein encodes a further three non-structural proteins (NSP) responsible for RNA editing (V and W proteins) or with an open reading frame (ORF/C protein). Below is shown the comparison between NiV and HeV proteins (see Tables 1 and 2).

**Table 1.** NiV proteins (adapted from source: Uniprot, accessed on 8 March 2023).

| Protein Names | Gene Names | Length (Amino Acids) |
|---|---|---|
| RNA–directed RNA polymerase L (Protein L) | L | 2244 |
| Non–structural protein V | P/V/C | 456 |
| Glycoprotein G | G | 602 |
| Fusion glycoprotein F0 (Protein F) | F | 546 |
| Matrix protein (Protein M) | M | 352 |
| Phosphoprotein (Protein P) | P/V/C | 709 |
| Nucleoprotein (Protein N) (Nucleocapsid protein) | N | 532 |
| Protein W | P/V/C | 450 |

**Table 2.** HeV Proteins (adapted from source: Uniprot, accessed on 13 March 2023).

| Protein Names | Gene Names | Length (Amino Acids) |
|---|---|---|
| Fusion glycoprotein F0 (Protein F) | F | 546 |
| RNA–directed RNA polymerase L (Protein L) | L | 2244 |
| Non–structural protein V | P/V/C | 457 |
| Phosphoprotein (Protein P) | P/V/C | 707 |
| Nucleoprotein (Protein N) (Nucleocapsid protein) | N | 532 |
| Matrix protein (Protein M) | M | 352 |
| Glycoprotein G | G | 604 |
| Protein C | P/V/C | 166 |
| Protein W | P/V/C | 448 |

Recent outbreaks of NiV indicate that genome sequencing is required for the NiV infection that has occurred so far in isolated outbreak samples [33,34]. In 2019, in a case study, it was indicated that mutations occurred within the NiV N gene that the sequencing database sites (nextstrain.org) do not currently display; although massive global surveillance during the COVID–19 pandemic on viral mutations in SARS-CoV-2 spike (S) protein domains and other potential pathogens does continue [33,34].

*3.2. Nipah Virus Viral Protein Factors*

Two viral membrane glycoproteins (see Table 1) above are considered to mediate NiV cellular attachment and include the haemagglutinin–neuraminidase (HN) proteins. These attach to the receptor binding protein (RBD) and are considered stalks shared amongst other *Paramyxoviridae* requiring fusion peptides (F) with NiV possessing a NiV G glycoprotein [35]. These two proteins are type II membrane glycoproteins composed of a cytoplasmic tail and transmembrane region attached to the viral envelope with a stalk and globular head. Insertion of the hydrophobic NiV F protein alongside the NiV G protein into the cellular plasma membrane is required for viral propagation. Nipah virus entry to host cells was described as utilising recombinant NiV N/NiV P interaction, that is crucial in disrupting NiV viral mRNA. This affects polymerase transcription and replicase activity through assembly and encapsulation into full-length genomic viral RNA [36,37]. Furthermore, the N-terminal protein of NiV N could be inhibited, as with SARS-CoV-2, where the conformation and binding shape of the unassembled NiV N protein affect cellular entry. Hendra virus protein homology gave early indications that IFN signaling inhibition occurred for both type I and type II IFNs (IFN–$\alpha$/IFN–$\gamma$), that usually facilitates viral elimination. In 2003, the HeV V protein was thought to accumulate intracellularly with specific signal transducer and activator of transcription (STAT) proteins. These include STAT1 and others below that can be dysregulated, preventing further IFN-induced signaling [38]. Therefore, the NiV N protein, through saturation of a binding domain within the NiV P protein (polymerase complex), could affect enzyme activity through transcriptase or replicase activity [36,39]. Moreover, NiV M protein involvement in viral budding is considered to affect cellular morphology [40]. Nevertheless, F peptides are required to bind and traverse the phospholipid bilayer, dependently or independently of cellular receptors.

Further clarification from Michael Weis' group in 2015 (Marburg, Germany) indicated the NiV F protein contains Y525RSL (a tyrosine signal) that is essential for NiV F protein uptake, proteolytic activation, and endocytosis [41]. In comparison, Protein C researched in vivo can cause more serious respiratory damage during NiV infection as studied in knockout experiments [42]. In 2003, utilising an in vitro model of recombinant Newcastle Disease virus (NDV), all three NiV proteins (NiV V, NiV W, and NiV C) were indicated as required for intracellular NiV replication [43]. Cellular regulation of NiV appears to clarify in vitro that p38 is also essential [44,45]. Whilst X-ray crystallography illustrated that NiV M protein binds to intracellular phosphatidylserine (PS) and soluble cellular phosphatidylinositol 4,5-bisphosphate (PI (4,5) $P_2$). These maintain and regulate intracellular NiV virion particle formation and egress through NiV M protein polymerization [46]. The larger NiV L protein function is that of a viral polymerase required for replication [47,48]. Viral NiV replication therefore occurs intracellularly through post-transcriptional modification, resulting in 5′ capping and methylation of RNA. Thereafter, NiV V protein is described as the virulence factor that may abrogate type I IFN responses intracellularly, as we discuss below [49].

*3.3. Receptor-Mediated Nipah Virus Entry*

Initially, it was clarified that two ephrin receptor types, B2 and B3 (EFNB2/ENFB3), are both encoded by EFNB genes are implicated in cellular host HeV and NiV infection [50]. In 2003, EFNB2 receptor stability was clarified as being induced by the NiV G protein preceding full receptor activation, similar to a viroporin [51]. Ephrin B2 is preferentially expressed on vascular endothelial cells, as well as alveolar type II epithelial cells and around host naïve T cells ($T_N$), with a known role in angiogenic tissue remodeling [52]. Moreover, according to current data, EFNB2 is constitutively expressed at higher levels in the brain, cerebral cortex, amygdala, basal ganglia, hypothalamus, and white matter. Ephrin B2 is conserved across species and could be a therapeutic target. It appears that two NiV proteins, NiV F and NiV G, co-stabilize at the EFNB2/EFNB3 receptor interface to facilitate viral RNA cellular entry [53]. Microbial and viral pathogens utilise protein glycosylation for host cell entry and immune response evasion. This can occur through either N-glycans or O-glycans affecting cellular protein binding and signaling at the cell membrane surface. Therefore, N-glycosylation in NiV infection was indicated by unique N-X-S/T motifs (asparagine (N), any amino acid apart from proline (X), serine (S), or threonine (T)). Currently, O-glycosylation of S/T residues remains unknown. Other Paramyxoviruses are affected by N-glycans in MeV infections that alter F protein-mediated fusion and processing. In addition, C–type lectins (e.g., galectin–1) bind to N and O glycans that inhibit infection but increase endothelial viral RNA entry [54,55]. This was investigated in 2015, when the disruption of eight predicted N-glycosylation sites was examined to find six (G2–G7), with one conserved between HeV and NiV in the stalk (G2) and five in the head domain (G3–G7) [56]. The EFNB2-predominant receptor is better characterised through histological analysis and is absent in other cellular subtypes, but other unknown receptors may exist [50,57]. As shown below, the NIV G protein appears to be stabilized as a trimer with NiV F/EFNB2 during cellular fusion. Intracellularly, NiV F is activated, endocytosed, and can be cleaved through proteases (e.g., cathepsin L), but remains active during viral replication and virion formation [58]. Given our discussion below and considering current immunogen development, it appears as though the NiV F protein may either be less immunogenic or shielded from immune cells during cellular infection.

*3.4. Cellular Immunomodulatory Properties of Nipah Virus Infection*

Bats have been described as having IFN modulatory properties that enable viral transmission. Reports appeared in 2012 describing bat viral reservoirs [59]. Other reviews document both OPXV infection and SARS-CoV-2 affecting host IFN cell synthesis. Comparatively, less is known about the role that IFN plays in the cellular regulation of viral infection. Factors that have been described in *P. vampyrus* include type I/III IFNs modulating IFN gene transcription through receptors; for example, different species may have different

specific type I IFN subtypes (e.g., IFN–α/β) or type I IFN receptors (e.g., IFNAR–7). Other species like *M. lucifugus* and *P. vampyrus* may have more of the other type I IFN subtypes (IFN–ω). Earlier articles are indicative of a predominant type I/III IFN regulatory role in both human and bat species that can eliminate viral infections [59]. In comparison, type II IFN (IFN–γ) can also be considered important for adaptive immune responses in cancer [60,61]. Within NiV disease, there are suggestions that both NIV W and NIV V both appear to regulate IFN–β production [62]. This could occur by inhibiting the proteolysis of the UBXN1 protein and suppressing both retinoic acid-inducible gene I (*RIG–I*) and a similar helicase receptor (MDA–5). These stabilize mitochondrial antiviral signaling protein complexes (MAVS and ubiquitin-binding UBA domain/UBX motif, UBXN1) are known to regulate the nuclear transcription factor (NF–κB) [62,63]. Therefore, as NiV W can influence NF–κB, the detailed role of other pattern recognition receptors (PRR) like Toll-like receptors (TLRs) that are present on cell membranes and vesicles during NiV infection is starting to emerge. There are ten known TLRs in humans.

Interferon modulatory properties affect intracellular viral propagation and modulate immune responses that are well characterised in humans but less so in bats. These include interferon regulatory factors (IRF3/7) and TLRs (TLR3/4/7/8/9) known to be affected by viral DNA/RNA host cell entry. Many other PRRs are considered better known in other pathologies [64]. Early in NiV research, NiV V and NiV W proteins were seen in vitro to selectively antagonize host cell IFN gene transcripts. In the context of NiV infection, TLR 7/9 were plausibly seen to be inhibited, thereby additionally reducing cellular IFN–α synthesis and induction through the IκB kinase (IKK–α) and IRF7 pathways by inhibiting NF–κB transcription [42,65]. Very recently, MAVS were thought to be critical alongside both TLR7/MyD88, that could be redundant during NiV infection rather than TLR3. These recent reports indicate the biological plausibility that TLR7 on immune cells could affect activated T cell subtypes known to be preferentially infected and that express high levels of EFNB2 during active NiV infection [66]. The amino terminals of NiV V and NiV W proteins may be causal in this, as the NiV V protein carboxyl terminus is known to be relevant to HPIV-II infection [43]. Recently, mRNA transcripts were noted during in vivo NiV immunogen experiments. These researchers noted upregulation of transcription factors (TFs) relevant to IFN responses (*IFIT2*, *GBP1*, *IFIT1*, *MX1*, *OASL*, and *IFIT3*). However, other immune response genes were also noted as upregulated (e.g., *FCAR*, *CLEC4E*, and *IL-8*) [67]. However, as NiV V and NiV W diffuse intracellularly to the nucleus, these may inhibit the janus kinase (JAK) and STAT protein pathways affecting the *IFN–β* promoter and IRF3. Furthermore, extracellular NiV W in return may antagonize TLR3 around plasma cell membranes and intracellular vesicles. Below are shown known cellular interactions during NiV infection and upregulated molecular gene transcripts observed in research so far in italics (see Figure 2).

Other RNA viruses can affect TLR3 through Toll-interleukin–1 receptor (TIR) domain-containing adaptor-inducing IFN–β (TRIFs) [68,69]. Recent reports are indicative that IRF1 also plays a role in IFN regulation in both monocytes and macrophages (Mφ), together with IRF7/9 coordinating TLR sensing of viral DNA/RNA [70,71]. Furthermore, IFN regulation appears to be regulated through two NiV proteins, NiV V and NiV M, that enhance the tripartite motif (TRIM6) post-translational protein degradation. This could impair the oligomerization and phosphorylation of an IκB kinase (IKK–ε) [72]. As noted above, IFN–β is required for myeloid cell differentiation and maturation of antigen-presenting cells (dendritic cells (DCs), monocytes, and Mφ), as well as regulating Mφ apoptosis [73,74]. Data indicates that TRIM6 is also constitutively expressed predominantly by neutrophils within the immune system (see Supplementary Materials). Furthermore, STAT proteins were examined through immunoprecipitation of haemagglutinin (HA) proteins tagged to NiV P, NiV V, and NiV W proteins. These were flanked by a peptide sequence (FLAG, denoted by the DYKDDDDK sequence). The NiV P protein was observed to have a conserved interaction with STAT1, STAT2, and STAT4, but specifically with amino acid residues (114–140) of the NiV P protein. Whilst STAT1/4, but more specifically the STAT1 SH2 domain can affect

IFN–β nuclear transcription. In addition, it was found that NiV P/V/W proteins could also antagonize STAT4 through N-terminal domains [75]. Moreover, within endothelial cells, STAT4 protein in vitro research indicates that IFN-α rather than the DC maturation cytokine IL-12 is required to activate STAT4 alongside the CD4 $T_H1$ cell response [76].

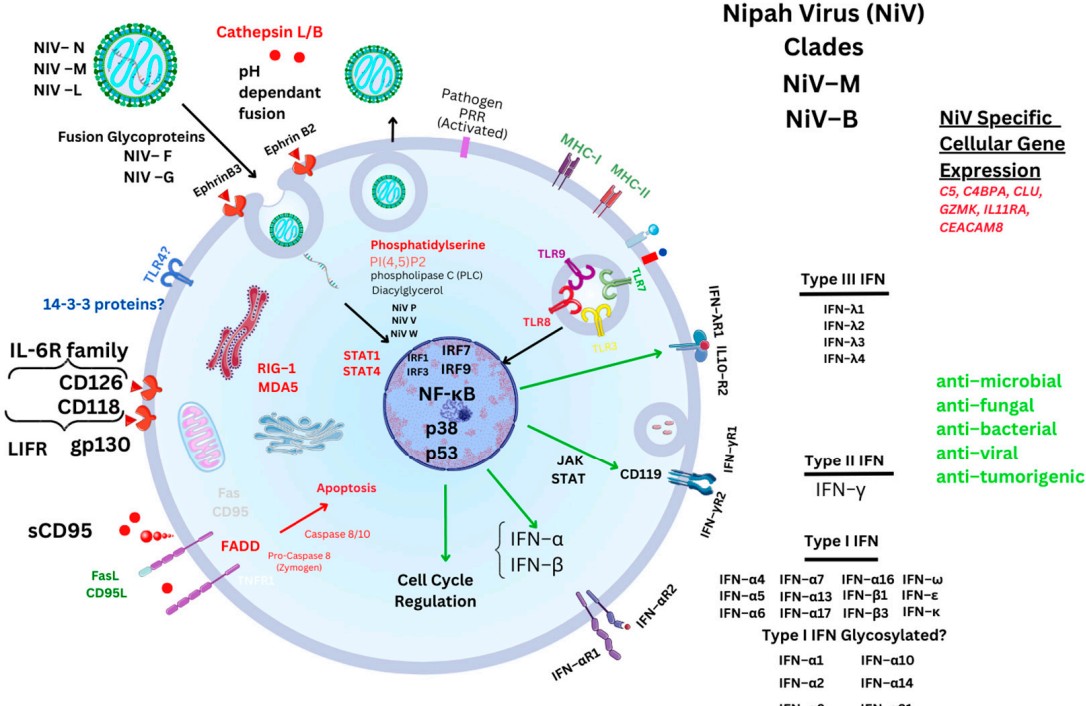

**Figure 2.** Nipah virus molecular and cellular regulation.

## 4. Current Therapeutics

### 4.1. Clinical Trials

Currently, no therapeutics are licensed for human NiV prophylactic infection or treatment. Development and clinical trials are ongoing with other immunogens. Currently, one vaccine is licensed for HeV in horses [77]. Research in vivo into future NiV immunogens indicates that therapeutics derived from the Vaccinia virus could be an effective attenuated viral vector expressing both NiV G and NiV F proteins [32]. The initial emergence of potential NiV vaccines therefore continued to be examined in vivo between 2004 and 2006 [78,79]. Presented below are the registered current clinical trials listed (see Table 3).

**Table 3.** Current Clinical Trials (Source: https://www.clinicaltrials.gov, accessed on 10 March 2023).

| NCT ID | Title | Locations |
|--------|-------|-----------|
| NCT04199169 | Safety and immunogenicity of a Nipah virus vaccine | OH, USA |
| NCT05178901 | Phase 1: evaluate safety and immunogenicity PHV02 | KS, USA |
| NCT05398796 | Dose-escalation evaluation of the safety and immunogenicity of mRNA-1215 | NIH, MD, USA |
| NCT01811784 | Community intervention to prevent Nipah spillover | Bangladesh |

### 4.2. Henipavirus Monoclonal Antibodies and Vaccines in Development

From 2015, newer therapeutics targeting *Henipaviruses* during outbreaks were suggestive that in vitro/in vivo Phase 1 studies established a monoclonal antibody (m102.4) that met safety, tolerability, and pharmacokinetic profiles to be suitable for use [80]. This followed up on 2013 research that indicated nAbs were produced against combined HeV/NiV

antigens utilising an epitope targeted against the EFNB2 G–H loop residue [81,82]. Several projects are currently undergoing evaluation for their promise for NiV therapy development. This research indicates that a soluble recombinant G protein subunit of HeV (HeVsG) may provide protection against known strains of *Henipaviruses* that are NiV–M and NiV–B, but also against HeV in four animal species. This HeVsG immunogen was initially licensed in 2015 as Equivac® HeV by Zoetis Inc., under regulatory authority by the Australian Pesticides and Veterinary Medicines Authority (APVMA) [83,84]. Similarly, in 2012, a chimpanzee Adenovirus replication-deficient vector (ChAdOx) was further investigated. Suggestions were that a limited or low antibody response may occur but would still provide protection against 2 NiV strains (NiV–B or NiV–M). This vector progressed through trials with Oxford University starting in 2012 [85]. Later, in conjunction with Vaccitech/Astra Zeneca from April 2020, ChAdOx was researched in phase 1/2/3 clinical trials and then under Emergency Use Authorization (EUA) with SARS-CoV-2 antigens to instead express NiV-specific antigens to effect the necessary innate and adaptive immunogenic response. Just prior, this vector was investigated for immunogenicity against tuberculosis, MERS–CoV, and Rift Valley fever [86–89]. Subsequently, research in vivo utilizing the ChAdOX vector encoding the NiV G RBD protein antigen observed that NiV N and NiV F proteins were comparatively not immunogenic, with binding antibodies at minimal concentrations [90]. In 2019, other researchers utilised immunoinformatics to further clarify NiV T cell epitopes. It was indicated that some were non-allergenic with 2 epitopes, LLFVFGPNL and KYKIKSNPL, that could retain antigenicity and bind to MHC class II alleles (HLA–DRB1*01:01 and HLA–DRB1*07:01) [91]. Furthermore, in silico studies are examining the design of self-amplifying mRNA vaccines against NiV infection [92]. More recently, the modified Vaccinia Ankara (MVA) vaccine is being engineered to express full-length NiV G protein or soluble NiV G protein [93]. In 2022, a recombinant Vesicular Stomatitis virus (rVSV) vector was trialled in vivo, with results suggesting that nAbs were essential for protection within 7 days [67]. Other reports indicate that a lipid nanocapsule protein (LNP)-encapsulated mRNA immunogen encoding the soluble HeV G protein is in development (see Table 3 and Supplementary Materials). Other recombinant MeV vaccine vectors are also in development that express NiV proteins [94].

### 4.3. Current Antiviral Therapeutics

After the 1998–1999 NiV outbreak, ribavirin was trialled, with results in 2001 indicative of a 36% reduction in mortality [95]. In 2014, a synthetic purine analogue, T–705, known as Favirapir (Avigan®) and authorized in Japan for Influenza treatment, began undergoing phase 2/3 trials in the USA/Europe [96]. Favirapir is a viral RNA-dependent RNA polymerase (RdRp) inhibitor observed in NiV–M infection in vivo to reduce viral antigen in the brain, lung, and spleen as observed in histopathological observation [97]. Before/after 2019, Remdesivir (GS–5734) was examined in phase 2 clinical trials, with its nucleoside analog seen to inhibit not just *Filoviruses*, but potentially also *Pneumoviruses*, and *Paramyxoviruses* similarly [98]. More recently, other host-specific inhibitors with broad-spectrum *Paramyxovirus* activity have been described. Undergoing further development are two compounds (ZHAWOC9045/ZHAWOC21026). Early indications are suggestive that these retain activity against not only NiV and MeV but also both RSV and HPIV (type V) through inhibiting the RdRp complex [99].

### 4.4. Diagnostic Methods and Future Prospects

Recently, during further assay development, literature indicates more specific assays have been developed. These will undoubtedly be useful and differentiate between viral infections, including NiV and HeV, utilising EFNB2 as the known ligand in combination with mAb–F20NiV-652022. At the time of writing this article [100], the affinity of the NiV G protein for EFNB2 and EFNB3 receptors was not yet conclusively known. Current proprietary lateral flow tests utilise detection of the NiV nucleocapsid (N) protein [101]. More recently, in 2022, 40 B and T cell epitopes were identified in molecular docking and

immunoinformatics research. These authors suggest that such epitopes could evoke immune cell release of a type II IFN (IFN–γ), interleukin–4 (IL–4), and interleukin–10 (IL–10). Molecular docking studies suggested that immune cell TLR4 and TLR8 pattern receptors could be crucial [102]. The results of this would be interesting to see. In our last publication, we compared immune system cellular markers of Mϕ during SARS-CoV-2 infection [17]. Macrophage polarization through activation and differentiation into M1 Mϕ or M2 Mϕ requires further clarification, and therefore recently other researchers have clarified such tissue-specific Mϕ [17,103]. However, above we discussed TLR3/7 inhibition with regards to NiV infection, and recent research recognizes that TLR agonists may hold promise in this regard [104].

## 5. Current Known Immunological Perspectives of Nipah Virus Infection

### 5.1. Background

In late 1999, initial reports of NiV infection indicated that lymphocyte count, platelet count, low serum sodium, and high aspartate aminotransferase concentration (each observed in five individuals) together with high CSF pleocytosis were serum factors that changed during NiV infection in humans. Antibody (IgM) concentrations in combination with PCR testing were considered indicative of NiV infection [105]. Below, we consider known serological studies in different hosts. These can vary between animals due to the characteristic differential expression of CD molecules or other proteins. Laboratories confirmed that initial enzyme-linked immunosorbent assays (ELISA) could be 98.4% specific for *Henipaviruses*. These require validation of the specificity of the utilised monoclonal antibodies before diagnostic or clinical consideration [106]. Little is known about the immunology of NiV infection. In India (2001), reports confirmed the same high mortality rates as before, with serological analysis available. Serology responses (IgG and IgM) in humans were detectable in 9/18, or 50% of NiV infections [18]. However, in vitro and porcine studies up to 2012 imply that NiV directly infects $CD4^-CD8^{hi/+}$ and $CD4^+CD8^{hi/+}$ T cells expressing CD6. Importantly, downregulation of IFN–α gene transcription can occur, that would in effect reduce IFN–α synthesis and secretion. At the 7-day timepoint, $CD4^+CD8^-$ T cells significantly reduce when human mortality can occur; these cells form a key part of adaptive immune systems [44]. Inhibition studies of CD6 illustrate that this cellular marker regulates and influences leukocyte CD4 T cell extravasation across endothelial cell membranes and the blood brain barrier. CD6 is also preferentially expressed by activated monocytes, epithelial cells, and neurons [44,107–109]. Other *Morbilliviridae*, such as MeV, utilize immune cell receptors like signaling lymphocyte activation molecules (SLAMF1/CD150) to replicate within the immune system and are expressed on T cells, B cells, natural killer (NK) cells, and DCs [110,111].

### 5.2. Analysis of Cellular Nipah Virus Immunogen Responses

Recent reports investigating potential future immunogens to NiV infection indicate that the rVSV immunogen expressing the NiV G glycoprotein protects all non–human primates from mortality within 7 days [112]. Interestingly, in this comparison, it was noted that gene expression of *MX1*, *OASL*, *IFI44*, *GBP1*, *IFIT2*, and *IFIT1* were key downregulated transcripts [113,114]. Within these interferon-regulating gene TFs, *IFI44* is notable as a shared interferon-inducible factor between viral and bacterial infections, including SARS-CoV-2, *Staphylococcus aureus*, but also Rheumatoid Arthritis (RA) [115]. Furthermore, more genes that affect adaptive immunity were noted as upregulated, including *CD96*, *KLRK1*, *KLRG1*, *KLRF1*, and *SH2D1A* [116,117]. These were interesting observations, as *KLRK1* and *KLRG1* are implicated in lung adenocarcinoma as potential prognostic markers. However, *KLRF1/KLRG1* can be associated with TNF/IFN reduction, which is suggested to occur from within the $CD4^+$ T cell compartment [118–120]. Additional upregulation occurred in survivors within NK cell compartments expressing the gene transcripts *KLRC3*, *KLRC2*, and *GZMM*. These three genes encode the NK cell receptor proteins NKG2E/NKG2H and NKG2C, together with the neutral serine protease granzyme M, that is stored in activated

NK and T cells (see Supplementary Materials). The CD96 receptor can affect extracellular and intracellular TFs, including T cell immunoreceptor and immunoreceptor tyrosine-based inhibitory motif domains (TIGIT), CD226; T-cell immunoglobulin and mucin-domain containing-3 domains (TIM–3); programmed death–ligand (PD–L1); cytotoxic T leukocyte antigen 4 (CTLA–4); and STAT3 [121]. Below are shown the current immune cell interactions during NiV infection, that are discussed further below (see Figure 3).

## Henipavirus Immune System Cell Interactions

**Figure 3.** Immune cell interactions.

*5.3. Current Implications and Immunological Research*

During the 2019 outbreaks of NiV infections (n = 18), further reports appeared of survivors (n = 2) that have a dominant cytotoxic T cell ($T_C$) response. This was indicated by the T cell marker CD8 and denoted by MHC class II molecule expression (HLA–DR$^+$) together with CD38$^+$ but without change in the $T_H$ cell compartment. In addition, both PD–1 and granzyme B were documented in individuals (n = 2) recovering from NiV infection with serology analyses. In these individuals, B cell plasmablasts matured with the secretion of specific and detectable IgG and IgM, peaking at 5 days [122]. Very recently (2020), more detailed information is known. In comparable in vivo research in cynomolgus monkeys, it was seen that during NiV infection, gene transcription could provide detail on NiV disease that could plausibly influence coagulation, cytolysis, cytokine production, and other pathways. These include complement genes (*C5/C4MPA*), granzymes (*GZMK*), the cytokine IL–11 receptor (*IL11RA*), clusterin (*CLU*), as well as carcinoembryonic antigen–related cell adhesion molecule 8 (*CEACAM8*) (see Supplementary Materials) [123]. This upregulated gene transcription was accompanied by decreased expression of one immune cell receptor, CD244. The CD244 protein is encoded by *SLAMF4,* which encodes signaling lymphocyte activation protein (SLAM) expressed by both NK and T cells that may restrict NK cell-driven MHC-restricted cytolysis. CD244 can also be expressed by monocytes, NK cells, and, at elevated levels, basophils. Other gene transcripts concurrently noted as upregulated (*CD79A*, *IRF8,* and *DEFA1)* that could also respectively affect B cell differentiation, IFN

transcription, and defensin protein production. In NiV survivors, upregulation of *SLAMF6*, SLAM-associated protein (*SH2D1A*), family tyrosine kinase *(FYN)*, and chemokine receptor (*CCR5*) was considered significant. Furthermore, T cell genes upregulated included *CD3 (D, G,* and *E)* and *CD8a*, whilst T box TF *(TBX2)* was downregulated in late disease. The role of the IL1R2 protein in the regulation of IL–1α and IL–1β synthesis remains unclear [124]. IL1R2 is composed of an extracellular domain composed of three glycosylated immunoglobulin (Ig)–like domains and lacks the intracellular TIR domain and can sequester IL–1α and IL–1β [125]. Regulation of IL–1 secretion, tumor necrosis factor (TNF–α), IL–6R, TNF receptor (TNFR1/2), and transforming growth factors (TGF–α) each occur during homeostatic immune cell responses utilizing adhesion molecules (e.g., L–selectin (CD62L) and ICAM–1) [126,127]. Differential effects can affect neutrophil chemotaxis [126,127]. Monocyte and Mϕ chemotaxis requires maturation cytokines and CD62L to cross endothelial cell membranes, as discussed in our last publication [17,128–130]. Comparatively, within viral pathologies, DC–SIGN (CD209) receptors can be predominant in myeloid lineages, as in HIV–1 and Zika virus, that facilitate infection of DC subsets [128–131]. More research would undoubtedly clarify whether these affect NiV cellular infection. In recent reports further examining the host immune response using the VSV vector against NiV, gene transcripts were observed for *CD40L*, *CD3E,* and *CD8A* alongside *CX3CR1*, *IL-21,* and *TCF7.* These are key regulators of NK and T cell function [67,132]. Key cytolytic gene transcripts for granzymes (*GZMA/L/B*), *granulysin*, *KLRC2,* and *KLRC3* were upregulated, of which some could potentially be synthesized by T$_C$ cells with others encoding C lectin receptors on NK cells or DCs [67]. With regards to NiV direct infection of APCs, it remains unclear as to whether this involves clear active replication, permissive infection, or latency as seen in other familial viruses (e.g., *Retroviridae, Coronaviridiae, Herpesviridae,* or *Orthopoxviridae*). Nipah virus was indicated as appearing to lack fusion ability with myeloid-derived cells, including DCs, monocytes, and Mϕ. This is consistent with the comparatively lower expression of EFNB2 or EFNB3 [50]. Reports are contradictory, although earlier research refers to interstitial DCs that develop through maturation into different phenotypes as anti-inflammatory, pro-inflammatory, or anti-tumorigenic [133]. Interstitial cells were originally defined in 1978 as distinguishable from tissue Mϕs histologically by high staining and expression of MHC type II molecules that present peptide antigens [134].

In 2013, it was clarified in vitro that four pro-inflammatory cytokines were produced within four days by DCs after NiV infection. These are IL–1β, TNF–α, IL–10, and CCL3, with reduced type I IFN synthesis. A concomitant upregulation of three immune cell markers (CD40, CD80, and CD86) occurred that affects the tolerogenic profile of DCs [135]. Of these proteins, CCL3 is known to be a chemoattractant for six immune cell types, including T cells, B cells, basophils, eosinophils, mast cells, and NK cells. Comprehensive recent reports document in vivo serum and immune cells up to day 14, describing respiratory infection of epithelial cells, endothelial cells, and CD68$^+$ Mϕ, but not pneumocytes expressing E–cadherin [135]. Therefore, this could be indicative of either pro-inflammatory or anti-inflammatory Mϕ or tumor-associated Mϕ not yet known [136]. However, as we discussed in our last article, immune cells can be described by characteristic CD markers and chemokines [17]. Pleiotropic cytokines and receptor families remain a focus for future research to elucidate signal transduction mechanisms. For example, the role of the IL–6 family of receptors, which includes the leukemia inhibitory factor receptor (LIFR) and ligands, remains under research and development. Recent discoveries implicate LIFR within the family of IL–6 receptor subunits as being regulatory in other pathologies. These plausibly may affect JAK and STAT3, but also mitogen-activated protein kinases (MAPK) and a serine threonine kinase (AKT/protein kinase B, PKB) that will be discussed further [17,137–140].

## 6. Discussion

Nipah virus and other BSL4 pathogens represent a present public health threat that have been comparatively recently discovered and isolated since 1994. Hendra virus characterization and Nipah virus isolation during sporadic laboratory culture developments

and outbreaks since 1999 are indicative of this. Comparatively, less is known about the role that IFN plays in immune cell regulation within Pteroptids, which therefore should be a key area of research. This would elucidate how zoonotic regulation occurs within different species of animals that possess differential cellular immune system markers. According to the WHO website, these are listed as unknown landscape, target product profiles, trial designs, and scientific consultations (see Supplementary Data). Infection with the Nipah virus remains a focus. Recent history, since its discovery in 1999, with pandemics caused by other viruses like Influenza (H1N1) and SARS-CoV-2 is suggestive that scientific rationale is required to contextualize the relevance of research. Risk factors for Nipah virus transmission are suggestive of epidermal contact, food transfer, and an unclear transmission rate during currently reported sporadic outbreaks [141–143]. However, comparisons with viral outbreaks in other herd animals are noteworthy. Recently, the H1N1 strain of *Orthomyxoviridae* (Influenza) was almost continuously recorded from 1974 to 2015. Other strains of Influenza (H5N1/H5N8) were noted as increasing in animal herds in recent reports. For example, in Europe, 2520 outbreaks of H5N1 Influenza occurred in animals in October 2021, resulting in 3867 dead birds [144,145]. Furthermore, H5N1 outbreaks were reported in mink farms indicated during the COVID–19 pandemic as potential amplifying host reservoirs. More recently, discussions have started occurring to ascertain the effectiveness of disinfecting the Nipah virus, with the prospect that this can be done within 8 min using quaternary ammonium compounds or ethanol [146,147]. Reports between 2006 and 2012 are indicative of bat Nipah virus antibody prevalence that ranges in *P. medius* at around 6–7 years [148]. In contrast, in humans, other *Paramyxoviruses* can cause serious disease, for example Measles, where real-world data estimates of mortality suggest that this remains greater than 100,000 individuals per year. Measles virus is considered to reduce the antibody repertoire produced by B cell plasmablasts from 73% to 11% [149,150]. The apparent reduction in type I IFN secretion seen with human NiV infection, specifically IFN–β, may regulate DC, monocyte, and Mϕ maturation through cellular apoptosis, whilst NiV potentially replicates within T cell compartments. The role that CD244 receptor down-regulation performs, which is expressed by NK cells, basophils, and eosinophils, remains unknown but could plausibly affect B and T cell signaling.

## 7. Conclusions

In conclusion, given the encouraging immunogen developments we have outlined above for cellular and current molecular immunological research, we hope this paper provides further clarity to clinicians, researchers, and academics around the world. There are similarities between the Measles virus and the Nipah virus during the technological evolution of the immunological field occurring between 1994 and 2023. Above, we have presented known facts regarding Nipah virus with regards to protein, cellular infection, and immunology affecting IFN regulation in context with research since 1999. All proteins, genes, and immune cell mechanisms that the NiV infection is considered to affect during a lethal infection are displayed. Given the concerns that other emerging viral infections post-COVID–19 pandemic may pose, it is prudent to provide a detailed analysis and evaluate known host immune system responses in context. However, it is apparently clearer now that, within different viral outbreaks, knowledge gaps exist that require development. For example, in earlier Measles virus infections, it can clearly be seen that there is a suggestion of biphasic adaptive immune response regulation coinciding with $T_H17$ cell characterization. Such variations during any host immune response to different infections result in the induction of differential cytokines and chemokines within crucial B and T cell compartments. Much remains unknown due to sporadic Nipah virus outbreaks with regards to other immune receptors, such as Toll-like receptors and other pattern-sensing receptors. In other non-viral infections, some clarity appears with the above potential factors, originally discovered in 1957. These are type I, II, and III IFNs that appear to be central regulators of nuclear transcription factors and antiviral activity through the innate and adaptive arms of the immune system. Further research into the latter would

undoubtedly clarify and develop knowledge about the relevance between pathologies and innate immune system signaling in the future.

**Supplementary Materials:** The following supporting information can be downloaded at: https://www.mdpi.com/article/10.3390/immuno3020011/s1, Figure S1: Historical structure perspectives of *Paramyxoviridae.*; Figure S2: Nipah virus molecular and cellular regulation. Figure S3: Immune cell interactions. Table S1: NiV proteins (adapted from source: Uniprot, accessed on 8 March 2023). Table S2: HeV Proteins (adapted from source: Uniprot, accessed on 13 March 2023). Cynopterus brachyotis (Lesser short-nosed fruit bat) | Taxonomy | UniProt, Eonycteris spelaea (Lesser dawn bat) | Taxonomy | UniProt, 9823 in UniProtKB search (390336) | UniProt; who_nipah_baseline_situation_analysis_27jan20188b415c76-8649-4e98-b18f-18c6134f3819.pdf; NIH Launches Clinical Trial of mRNA Nipah Virus Vaccine | NIH: National Institute of Allergy and Infectious Diseases; Biohazard Levels–StatPearls–NCBI Bookshelf (nih.gov); Animals in Science Regulation Unit–GOV.UK (www.gov.uk); Laws, Regulations, and Standards Governing Research with Animals | CDC Laboratory Portal | CDC; EU regulations on animal research | EARA; Nipah Virus (who.int); Immune cell–TRIM6–The Human Protein Atlas: KLRK1 killer cell lectin like receptor K1 [Homo sapiens (human)]–Gene–NCBI (nih.gov): Brain tissue expression of IL11RA–Summary–The Human Protein Atlas; Encephalitis–Symptoms and causes–Mayo Clinic; Biosafety in Microbiological and Biomedical Laboratories (BMBL) 6th Edition | CDC Laboratory Portal | CDC.

**Author Contributions:** Original manuscript draft written by: B.B.; conceptualisation by: B.B., C.I.; formal analysis by: B.B., T.G., T.-N.C., C.I., S.A.A.-S., C.Y.L. and I.F.; data curation by: B.B., I.F., T.-N.C., C.Y.L., C.I., T.G. and J.A.; writing—original draft preparation by: B.B., T.G., T.-N.C. and I.F.; writing—reviewing and editing by: B.B., T.G., I.F., S.A.A.-S., C.Y.L., C.I., J.A. and T.-N.C.; software: B.B. and I.F.; visualization: B.B. and I.F.; supervision by B.B., I.F., T.G. and S.A.A.-S.; validation: B.B., T.G., I.F., S.A.A.-S., C.Y.L., C.I., J.A. and T.-N.C. All authors have read and agreed to the published version of the manuscript.

**Funding:** This research received no external funding.

**Institutional Review Board Statement:** Not applicable.

**Informed Consent Statement:** Written informed consent has been obtained from the patient(s) to publish this paper. Research contained within, refer to original citation(s) for further details.

**Data Availability Statement:** Above and upon request to the corresponding author or from sources contained within the text.

**Conflicts of Interest:** The authors declare no conflict of interest.

## Abbreviations

Signal Transducer and Activator of Transcription (STAT1/2/3), Interferon-induced GTP-binding protein (MX1/MX2), Interferon gamma-induced protein (IP10), OAS ($2'$-$5'$-oligoadenylate synthetase 1 OAS1/2/3), and Interferon-inducible protein (IFI). For other protein abbreviations not listed, see www.reactome.org (accessed on 14 February 2023).

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
