# Peer review of "Immunopathogenesis of Nipah Virus Infection and Associated Immune Responses"

_2673-5601, doi:10.3390/immuno3020011_

Round 1

Reviewer 1 Report

The authors submit a manuscript that addresses the modulation of adaptive immunity by Nipah virus. This virus is highly pathogenic and it is the authors' goal to investigate the pandemic potential since mortality is very high and there is no vaccine yet.

This is a review that includes nearly 400 publications from the last 25 years. In the introduction, the virus is introduced. This is followed by pathogenesis (clinic, proteins and genes, receptors utilized for infection, and immunomodulation). In the latter aspect, IFN-beta in particular seems relevant, not surprisingly. Epidemiology and a historical overview follow, furthermore therapies. The heading 4.4 should probably be 5., here it is about diagnostics. After a short discussion, the authors conclude that there is still much to be done.

Author Response

Thank you for your comments. 

We have made some amendments on the submission and section numbers with separate figures to elucidate known research so far. 

Reviewer 2 Report

The manuscript “Modulation of Adaptive Immunity: A Profile of Nipah Virus 2 and Associated Immunity Factors” presents very detailed information on the biology of Nipah virus 2. The work raises interesting aspects about the Clinical manifestations and diagnosis of this infection, as well as Virus Cellular Entry and Mechanisms, Receptor mediated Nipah Virus Entry, and Diagnostic Methods and Future Prospects as well, . However, the work presents historical aspects, not only of the infection by this virus but by other infectious processes that are very detailed and make this review not allow to identify in a relevant way the objective of explaining adaptive immunity and the analysis of immunogenic potentials. that should be considered in the generation of preventive measures against this infection. For example, both the summary and the introduction and topics related to the Epidemiology and Historical Overview, various aspects are raised that do not allow identifying the development of the main topic, the work in these conditions is only a compilation of information (v.gr. Current Implications and Immunological Research, clinical trials among others), so it would be highly relevant before this work is accepted for publication that the authors reconsider the length of the manuscript and the excess of information. It would be interesting if the authors proposed more than one drawing or scheme to suggest the immune response mechanisms (innate and adaptive) that are significantly affected by this virus. The work is interesting but it presents too much information that makes it difficult to analyze the relevant aspects and the objective is therefore confusing. The work must be better directed at the main objective, which is the modulation of immunity.

Figure 1 is too complicated to follow

Author Response

Thank you for your review.

You commented that:

Figure 1 is too complicated to follow

The work must be better directed at the main objective, which is the modulation of immunity so we have separated the images to the best of our ability according to known molecular, gene and immune cell knowledge so far. 

Additionally the title of paper has been amended to reflect profile of Nipah virus. 

Yes I agree the length of article is quite long and detailed. So we have removed the methodology section and re-worded the content to reduce this further.

Additional changes have been made with updated research since submission. 

Reviewer 3 Report

The article by Brown et al, discusses the Nipah virus associated immunity and current status of diagnostics and therapeutics. This review provides a good coverage of the literature and encompasses multiple important areas on Nipah virus biology. But the manuscript suffers from poor English language and major editing errors throughout which reduces comprehensibility of the article.

Major Comments

1.      Significant grammatical errors throughout the article. Major editing and corrections are required.

2.      Abstract needs to be more concise.

3.      Methodology is written before introduction.

4.      Significant overlap between subsections in section 5 with preceding sections. Also topics covered in this section should come before the discussion on current therapeutics.

Minor Comments:

1.      L23: What classification is indicated in the class 4?

2.      L28: There have been more than 2 pandemics in the last 2 centuries (e.g. H1N1 pandemic of 2009).

3.      L26 to 42: Major grammatical corrections required.

4.      L49: What is meant by “1331 reports listed under “Nipah” with 105 listed reports”?

5.      L116: Scientific names must be italicized.

6.      Main heading is missing for Section 5

Author Response

Thank you for your review and comments. 

In relation to major comments.

 1. Significant grammatical errors throughout the article. Major editing and corrections are required. We have correct most of these and reduced abstract.

2. Methodology has been removed according to other review. 

3. Significant overlap between subsections in section 5 with preceding sections. Also topics covered in this section should come before the discussion on current therapeutics. We agree and some of this has been changed. 

 4.   Significant overlap between subsections in section 5 with preceding sections. Also topics covered in this section should come before the discussion on current therapeutics. There is overlap and some has been changed. 

  1. L23: What classification is indicated in the class 4? Amended
  2. L28: There have been more than 2 pandemics in the last 2 centuries (e.g. H1N1 pandemic of 2009). Amended
  3. L26 to 42: Major grammatical corrections required. Amended
  4. L49: What is meant by “1331 reports listed under “Nipah” with 105 listed reports”? Removed
  5. L116: Scientific names must be italicized. Done
  6. Main heading is missing for Section 5. Done

Reviewer 4 Report

In this manuscript, Brown et al. has provided an extensive review of pathogenesis, epidemiology and therapeutic treatment of Nipah virus, with a focus on changes in immune signatures upon Nipah virus infection. Overall, the review is comprehensive and has covered multiple aspects of Nipah virus in both pathogenesis and molecular mechanisms. However, some parts of the manuscript need to be changed and re-organized. The language also needs editing.

Specific comments:

1) The abstract is too long and needs to be edited. The background information and some details need to be removed. The abstract should be a summary of the main topics / content that will be reviewed in the manuscript.

2) The ‘Methodology’ section is not needed for review.

3) “2.5 Immunomodulatory Properties of Nipah Virus Infection” can be combined with the Section 5.

4) Section 5 doesn’t have a subtitle for the section.

5) The title indicates the manuscript is mainly focused on the immune factors associated with Nipah virus but the review also covers many other aspects of Nipah virus infection. It’s better to have a more general title for the manuscript.

Author Response

Thank you for your time to review our article. 

Specific comments:

1) The abstract is too long and needs to be edited. The background information and some details need to be removed. The abstract should be a summary of the main topics / content that will be reviewed in the manuscript.

2) The ‘Methodology’ section is not needed for review. 

Noted and removed. 

3) “2.5 Immunomodulatory Properties of Nipah Virus Infection” can be combined with the Section 5. Amended title. 

4) Section 5 doesn’t have a subtitle for the section. Changed. 

5) The title indicates the manuscript is mainly focused on the immune factors associated with Nipah virus but the review also covers many other aspects of Nipah virus infection. It’s better to have a more general title for the manuscript. Noted and changed to profile. 

Round 2

Reviewer 2 Report

Unfortunately, the presentation of this work continues to be confusing, for example the title in which it is intended to identify A Profile of Nipah Virus Infection, as well as Associated Molecular and Protein Immunity Factors, however, this second objective is not identified in the development of the manuscript, since it is not clear to which molecular associated proteins it refers. In general, the objective of this work is very ambitious since it is intended to make an anthology related to the disease caused by this virus at the same time.

Author Response

Thank you for your time to review our article. 

The title can be amended to reflect your comment to:

A Profile of Nipah Virus Infection: Current Insights into Genetic and Adaptive Immune System Factors up to 2023

Reviewer 3 Report

I thank the authors for the thorough revision and I find the present version satisfying. Minor editing of language is required.

Author Response

Thank for you for taking the time to thoroughly review our manuscript and your comments.